# OpenReview forum: "Computational Barriers to Filtering for AI Alignment"
_ICLR.cc/2026/Conference — ICLR 2026 Poster_

### Official Review · Reviewer_938F · 2025-10-27

**Soundness:** 2
**Presentation:** 2
**Contribution:** 2
**Rating:** 2
**Confidence:** 4

**Summary:**

The paper considers two natural methods of filtering LLM content to prevent the generation of unsafe information—input and output filtering. The core result is that, given the existence of time-lock puzzles, AI safety cannot be achieved through the use of external filters: in other words, white-box model access is required to ensure LLM alignment. The authors show this by first demonstrating that there are LLMs for which adversarial prompts that are cryptographically indistinguishable from benign prompts can be efficiently generated; for such LLMs, no efficient input filter exists. Moreover, they show conversely that there also exist LLMs that can generate outputs that are indistinguishable from benign outputs but which nevertheless have harmful consequences.

**Strengths:**

- The connections to cryptographic hardness is interesting.
- Formalizing the difficulty of input/output filtering is a potentially useful step towards understanding the difficulties facing LLM alignment.
- It is interesting that their result for output filtering holds for filters that are stronger than the base LLM

**Weaknesses:**

- Consider shortening the title
- I’m not sure what scientific value figure 1 provides (perhaps worth keeping it for talks but not for a scientific paper)
- The structure of the paper is very unusual for an ML conference paper and makes it hard to read. The introduction is very unusually long. It is followed by a shorter technical section and a short reflection in the end. For example, the related work and the background is scattered throughout. It is very hard for me to accept the paper in its current form based on this alone.
- The setup in section 1.1 is quite dense and hard to read through.
- Regarding the experiments in section 1.3 are: The theoretical results concern, respectively, the existence of LLMs for which no filter can filter harmful from benign inputs, and the existence of an LLM that can generate harmful outputs that are indistinguishable to any efficient output filter from those of a reference LLM. Why does it matter that these particular filters and particular LLMs can be bypassed? It just means some filters don't work.
- The theoretical results consider "worst-case" LLMs, unclear how this lines up with empirical concerns, especially since their experiment doesn't seem that relevant to the theory.

**Questions:**

- I do not understand the philosophical perspective. What do "intelligence" and "judgement" mean here, and why do these results prove these notions cannot be separated?
- Why would one restrict themselves to only input/output filters when a company can filter intermediate embeddings as well?
- Do the theoretical results extend to intermediate embedding filters as well?
- Could the authors comment on how is their paper related to “Position: Fundamental Limitations of LLM Censorship Necessitate New Approaches”?
- How do the experiments in section 1.3 relate to the theoretical results?

---

> ### Author Response · Authors · 2025-11-21
>
> W1/2/3:
> We will shorten the title to "The Computational Cost of Filtering for AI Alignment" and change the figure to visualize the pipeline more scientifically. We will also change paper structure through renumbering (which makes the intro shorter), remark on the philosophical perspective in one small subsection only, and move the related literature part from the appendix to the main paper. We hope that including the Technical Overview in the body of the paper and proofs in the Appendix makes the paper easier to understand.
>
> W4:
> We will refine this section in order to make it more verbose and thus clearer. Do you have concrete suggestions?
>
> W5/6:
> See our answers to Question 5.
>
> Q1:
> After reading through the reviewer’s comments we understand that the philosophical discussion about intelligence and judgement was overemphasized. What we meant by “intelligence” and “judgment” was the following:
> - intelligence: for a machine whose sole purpose is answering prompts its intelligence is
> the quality of its output. Crucially, this machine always answers. We think of the LLM
> as the intelligence part.
> - judgement: for a machine, whose purpose is to evaluate harmfulness of prompts (and
> outputs) its judgement is the quality of its estimates. We think of the filters as the
> judgement part. \
>
> By impossibility of separating these two parts we mean that *"If the intelligence part (the
> LLM) is not aligned it’s impossible to align it with an external judging filter."*
>
> Q2/3: Our result relies on the running-time asymmetry between the filter and the LLM. A filter that rejects based on the embedding of an intermediate layer can be seen having fewer computational resources than the full network. It is thus
> possible to imagine that an extension of our result to such filters is possible. However, in our
> current proof the malicious LLM M′ has a special mode that activates harmful behavior and
> this might be detected by filters having access to internal representations. To address it one
> can imagine applying a cryptographic tool called indistinguishability obfuscation (which would
> effectively realize the black-box assumption) to hide the existence of such a malicious mode.
>
> Our additional experiments (see point (c) under General Practicality comment) demonstrate that the
> model’s internal alignment mechanisms can be circumvented. This is relevant because these
> mechanisms operate with implicit access to intermediate embeddings. The fact that they can
> be bypassed provides evidence that extending our theoretical result to intermediate-embedding
> filters may be feasible. While formally proving such an extension is outside the current paper’s
> scope, the empirical signal points in that direction. We thank the reviewer for suggesting this extension.
>
> **Due to character limit we will continue answering the remaining questions in the next comment below**

---

> > ### Author Response · Authors · 2025-11-21
> >
> > **Continuation of answering first rebuttal round**
> >
> > Q4:
> > This is indeed a very relevant paper (called [2] in the following), which we will add to our section on related works with a comparison. The work in [2] and our Theorem 2 consider a similar problem: the impossibility of detecting harmful outputs of LLMs (so called censorship). However, there are several differences:
> > 1. First, one of the attacks described in [2] relies on the LLM and the attacker "sharing a secret" (e.g., a function $g^{-1}$ in [2]) that the censoring party does not know. The LLM encodes its answer by applying g, and then the attacker decodes it by computing $g^{-1}$.
> >         In our time-lock construction, the attacker and the LLM *do not* need to share a secret key, i.e., they don't collude, and yet it's possible for the LLM to "lock" its answer and for the attacker to unlock it by investing extra time, whereas the output filter (analogously censor) will not be able to unlock it. To achieve this result, we needed a cryptographic assumption: the existence of time-lock puzzles where unlocking provably requires extra time. This enables us then to reach the *main takeaway message of our paper:* alignment is impossible if asymmetric resources are dedicated to alignment/censorship and to the attack.
> > 2.  The authors of [2] suggest a method to circumvent censorship, which is to split a harmful prompt into sub-prompts (so-called mosaic prompts) such that each looks benign, but aggregating the answers to each one allows the creation of an answer to the original harmful prompt. A major contribution of our paper is to show a *provable* method (under cryptographic assumptions) to avoid censorship of harmful prompts. Namely, we show how to transform/encode the harmful prompt into a benign prompt using (a special type of time-locked) steganography which cannot be decoded unless sufficient time has been invested. As explained in the paper, the idea behind time-locked steganography is to construct time-lock puzzles which are indistinguishable from random and embed them in the randomness of a small LLM which transforms harmful prompts into a distribution of prompts which is provably indistinguishable from a benign distribution of prompts.
> > 3. In the appendix of [2] candidates of syntactic censorship filters are proposed that try to detect suspiciously looking outputs. Based on our steganographic time-lock puzzle idea we *provably* show that no syntactic filter will work in the worst case.
> >         Thus, we are hiding a harmful prompt/output in plain sight.
> >         It might be useful to mention that this process is similar to prompt injection and prompt smuggling, where a harmful prompt is hidden in invisible Unicode characters, emojis, metadata, or embedded directly in pixels of an image, but our construction is provable.
> >  Summarizing the differences in one sentence, our method is *provable* and *does not* require collusion between the attacker and the LLM.
> >
> > Q5:
> > As explained in our general comment above, we performed more experiments that are more directly related to our construction.
> > To clarify the experiments in the paper: they simply show that filtering is already hard in practice as state-of-the-art filters are easily bypassed. Although these failures stem from different mechanisms than our theoretical constructions, both highlight the same core issue: attackers can craft prompts that look harmless to a filter but still elicit harmful model behavior. Our theoretical results formalize this in the worst case by showing that, under cryptographic hardness assumptions, no efficient filter can reliably prevent such attacks.

---

> > > ### Comment · Reviewer_938F · 2025-11-24
> > >
> > > I appreciate the authors’ thoughtful response to my comments.
> > >
> > > Could the authors compare their work with “GPT-4 Is Too Smart To Be Safe: Stealthy Chat with LLMs via Cipher” and clarify the theoretical and empirical distinctions between the two works?
> > >
> > > Since ICLR allows authors to revise their manuscripts during the rebuttal period, I would expect such clarifications and structural improvements to be reflected in the updated paper. In that context, it would seem premature to change the score prior to seeing concrete revisions, particularly given the paper’s unconventional structure.

---

> > > > ### Author Response · Authors · 2025-11-27
> > > >
> > > > We uploaded the revised manuscript with the changes you proposed. Please see the general comment “summary of revision”, in which we summarize all the changes we made.
> > > >
> > > > Concerning how we differ to “GPT-4 Is Too Smart To Be Safe: Stealthy Chat with LLMs via Cipher”:
> > > > They use a purely empirical approach to demonstrate that current LLMs can be jailbroken using encryptions
> > > > (Caesar, Morse, ASCII), where the entire conversation occurs in the cipher (including the outputs). This leads them to conclude that cipher data should be included in the safety training as a defense. In contrast, the main contribution of our work provides formal impossibility results: we prove under cryptographic assumptions that filtering-based alignment faces fundamental computational barriers that cannot be overcome through better training data. Our empirical setup also differs: we use time-lock puzzles where the model decodes encrypted jailbreaks and responds in natural language.
> > > >
> > > > Again, we added this comparison to the related literature section (section 6) in our revised paper.

---

> > > > > ### Comment · Reviewer_938F · 2025-11-28
> > > > >
> > > > > I appreciate the authors’ reply. I would like to raise my score to weak accept (6), but I am currently unable to do so due to OpenReview’s interface limitations.

---

### Official Review · Reviewer_xvj9 · 2025-10-27

**Soundness:** 3
**Presentation:** 1
**Contribution:** 3
**Rating:** 6
**Confidence:** 1

**Summary:**

This paper investigates the fundamental limits of achieving AI alignment through external filtering mechanisms. The authors focus on two natural points of intervention: filtering the input prompt before it reaches a Large Language Model (LLM) and filtering the output after generation. The core contribution is demonstrating, under standard cryptographic hardness assumptions (specifically the existence of Time-Lock Puzzles and One-Way Functions), that both input-prompt and output filtering can be computationally intractable.
hese results also extend to more expressive "mitigation filters" that can modify prompts or outputs. The authors conclude that an aligned AI system’s intelligence (the LLM internals) cannot be practically separated from its judgment (the external filters), necessitating "internal" alignment solutions. Empirical results with real-world filters (Llama Guard, Shield Gemma) are presented to support the theoretical claims.

**Strengths:**

The paper addresses a highly relevant and fundamental problem in AI safety and alignment, offering strong theoretical grounding for existing empirical challenges like "jailbreaking."

Novelty and Significance: The use of computational complexity theory and cryptographic hardness assumptions (Time-Lock Puzzles, OWFs) to model and prove the limits of alignment is a highly original and significant contribution. This elevates the discussion on alignment barriers from empirical observation to theoretical impossibility under standard assumptions.

Strong Theoretical Claims: The core theorems (Theorem 1 on input filtering and Theorem 2 on output filtering) provide separation results that are robust and compelling. The explicit requirement that the filter be computationally weaker than the LLM in Theorem 1 is a realistic constraint for many practical black-box or proprietary LLM deployments.

Comprehensive Scope: The paper doesn't stop at simple detection filters but also considers mitigation filters (Section 1.4) and explores scenarios involving shared secrets or public keys (Section 1.5), offering a well-rounded analysis of potential external defenses.

Connecting Theory to Practice: The inclusion of empirical evidence (Table 1) using state-of-the-art safety filters (Llama Guard, Shield Gemma) adds concrete, real-world relevance to the abstract theoretical findings. This successfully bridges the gap between the constructed, cryptographically-enabled LLMs and the observable failure modes of current models.

**Weaknesses:**

While the theoretical framework is strong, several aspects of the presentation, technical rigor, and scope could be improved for an ICLR audience.
* Lack of experimental support is the most critical factor. Otherwise, I think a position paper would be more suitable
* Technical Sketch Insufficient (Section 2): The high-level technical overview of the construction is abstract. Given that the entire proof hinges on the construction of the indistinguishable pair (G,G‘) and the function g, the key challenges mentioned—recovering n from (h,h(n)), the issue of multiple inverses, and using hardcore bits—should be explained with more detail or a more concrete example to demonstrate the Recoverable-Randomness Sampling mechanism. It is difficult to fully assess the proof's validity.

**Questions:**

NA

---

> ### Author Response · Authors · 2025-11-21
>
> **Weakness 1**: \
> Please see the general comment about the practicality of our results.
>
> **Weakness 2**: \
> We will give full details of the construction in the body of the paper.

---

### Official Review · Reviewer_JDRh · 2025-10-30

**Soundness:** 3
**Presentation:** 2
**Contribution:** 3
**Rating:** 6
**Confidence:** 2

**Summary:**

This paper examines the computational limits of achieving AI alignment through filtering techniques. The authors show that, under standard cryptographic assumptions, both input filtering (blocking harmful prompts) and output filtering (blocking harmful responses) face fundamental computational barriers. They construct examples of language models where no efficient filter can distinguish adversarial prompts or reliably detect harmful outputs. Even when filters are allowed to modify prompts or outputs instead of rejecting them outright, these more flexible strategies remain computationally constrained. Overall, the paper demonstrates that filter-based alignment methods are theoretically inadequate, emphasizing the need for a deeper understanding of the computational hardness behind AI alignment and for designing more robust regulatory and technical safeguards.

**Strengths:**

The analytical perspective of the paper is interesting.

**Weaknesses:**

1. The results rely on cryptographic hardness assumptions that, while widely accepted, remain unproven. If these assumptions were invalidated, the impossibility results would no longer hold.
2. The theoretical LLMs used in proofs involve contrived adversarial mechanism that may not fully reflect real-world systems. The gap between worst-case theoretical constructs and practical AI behavior limits its applicability.

**Questions:**

N/A

---

> ### Author Response · Authors · 2025-11-21
>
> **Weakness 1:** \
>
> Yes, it is correct. However, the practice of assuming well tested conjectures is standard in cryptography.
>         Indeed, the security of all private communication on the internet relies on such conjectures.
>         Thus, we believe it is valid to rely on similar type of assumptions in the context of AI Alignment.
>
> **Weakness 2**: \
> Please see the general comment about the practicality of our results.

---

### Official Review · Reviewer_Dtav · 2025-11-01

**Soundness:** 3
**Presentation:** 2
**Contribution:** 3
**Rating:** 6
**Confidence:** 1

**Summary:**

The authors study the problem of using external filters (either on prompts or on outputs) to prevent harmful behaviour from LLMs. They show, under standard cryptographic hardness assumptions, that there exist models for which no efficient prompt‐filter can reliably distinguish safe from adversarial prompts (i.e., adversarial prompts that trigger harmful behaviour are computationally indistinguishable from benign prompts). They also show that, in a natural setting, output filtering is computationally intractable: even if you observe the output, deciding whether it’s harmful or not cannot in general be efficiently done for certain models. They further explore relaxed mitigation strategies (weaker filter models) and demonstrate additional computational barriers.

**Strengths:**

A definitely novel investigation on the limitation of the (light-weighted) filter methods, based on the cryptographic argument.

It assesses the worst-case limitation of the filter based methods.

**Weaknesses:**

Practical relevance of Theorems 1 and 2 is limited, due to the assumption of the efficiency of the filter. It is not clear how much the filter needs to be powerful not to be considered as an “efficient filter.”

The claim of Theorem 2 is too informal to draw a meaningful message from it. Does the statement outputs of M′ are judged as harmful by H′” mean the outputs of M’ given arbitrary inputs are judged as harmful? Does the statement “​​no efficient output filter can distinguish the outputs generated by M′ from outputs of M.” mean one can not distinguish between the output of these two models for any input prompt?

The authors claim “The filter should be more efficient than the LLM; otherwise, we can ignore the given LLM and focus solely on training a new one from scratch, using it in place of the filter.” However, this may not always be acceptable. Because the safety guarantee is very important, it is natural to have more powerful filter models that are specialized for safety filtering at the cost of usefulness (such as guardian models).

**Questions:**

Please answer the above comments in the weaknesses section.

---

> ### Author Response · Authors · 2025-11-21
>
> **Weakness 1:** \
> When designing any AI system one needs to decide on the balance between the amount of resources devoted to capability and the resources devoted to alignment/robustness/safety. Our results essentially imply that filtering for alignment can not be done (in a black-box manner) using much fewer resources than used for capability (LLM). We believe this to be an important take away message, given the possible harm that can be caused by unaligned LLMs especially as we may be heading towards AGI.
>
>
> **Weakness 2**: \
> We thank the reviewer for pointing this ambiguity. The formal statement is in Theorem 10 (Appendix), and as always, simplifications need to be made when stating theorems informally.
>
> $M$ is always run on an empty prompt.The theorem shows that even in this restrictive setting there exists $M'$ such that, for every harmful prompt $m$, $M'(m)$ produces harmful outputs but $M'(m)$'s outputs are indistinguishable from outputs of $M()$ ($M$ run on empty prompt). The assumption that $M$ is run on an empty prompt could be replaced with any other fixed prompt. We will state the theorem more clearly in the final version.
>
> More formally, we assume the existence of a set of harmful prompts $m \in S$ for $M$.
> Then, the answers to two of your questions are:
>  - The statement "outputs of $M'$ are judged as harmful by $\mathcal{H}'$" means that for every $m \in S$, $M'(m)$ is harmful.
> - The statement "no efficient output filter can distinguish the outputs generated by $M′$ from outputs of $M$" means no filter can distinguish outputs of $M()$ from $M'(m)$ for every $m \in S$.
>
> **Weakness 3**: \
> We agree with you that safety is paramount and a lot of resources should be invested towards it.
>   Our result is essentially a proof that this must be done to achieve alignment.
>   We remark that what we see in practice however is that most guard models are lightweight LLMs with far fewer parameters than the main deployed models. For example, the largest and newest Llama Guard 4 model has around 12B parameters, whereas models like GPT-5 or Gemini are significantly larger. This may be partially due to the fact that from a user-experience standpoint, using very large guard models would introduce substantial latency and cost, making them undesirable.

---

> > ### Comment · Reviewer_Dtav · 2025-11-26
> >
> > Thank you for the response. However, the paper doesn't seem to be updated. Do you plan to revise the paper, and how?
> >
> > For the first weakness, I understand your point. However, at the same time, the informal, qualitative statement itself looks rather trivial, as it is very natural that a filter with a limited capacity can not filter out harmful prompts. To make the contribution clear, I feel that one should provide a quantitative assessment.
> >
> > Nearly the same applies to the second weakness. It is hard to derive a meaningful takeaway from the current statement, as mentioned in my original comment. The theorem statement as well as the explanation of the theorem must be restated.
> >
> > For the third point, the response was satisfactory. This must be clarified already in the introduction or so.

---

> > > ### Author Response · Authors · 2025-12-03
> > >
> > > We have uploaded an updated version of the paper that addresses all the points raised.
> > >
> > > **Regarding the first and second weakness:** We have revised the statement of Theorem 2 in the new version to explicitly specify the input prompt on which model M is executed. We also highlight that the formal version of Theorem 2 (Theorem 10) provides **quantitative and explicit specifications** of the running times for M, M', and the filters.
> > >
> > > We also would like to emphasize a key distinction between Theorems 1 and 2: unlike Theorem 1, Theorem 2 demonstrates that the output filter may run longer than the LLM M itself. The theorem holds under the natural assumption that the harmfulness oracle (the environment or downstream user) operates with substantially more computational time than the filter—specifically, sufficient time to unlock the time-lock puzzle prepared by the LLM. We believe this quantitative characterization directly addresses the weaknesses raised.
> > >
> > > **Regarding the third weakness:** We have incorporated the clarification you found satisfactory into the revised paper (see Section 2.1).

---

### Author Response · Authors · 2025-11-21
**Practical relevance**

Several reviewers asked whether our result is relevant to practice.

Recall that we *proved* (under cryptographic assumptions) the impossibility of alignment by filters whose runtime is significantly faster than the LLM, by demonstrating a cryptographic method to encode/hide harmful prompts that can be decoded and executed by LLMs but is impossible to decode (and even distinguish from a benign prompt) by fast filters.

To show the significance of our attack, we ran a series of experiments. In the first experiment of the paper (Table 1), we use natural language jailbreak prompts y, which try to elicit a harmful response. To tailor our experiments more to our theoretical results, we performed additional experiments using a timed-release mechanism inspired by our construction: we create an attack z which first tasks the model to decode E(y) using a cipher C, where E(y) is replaced by an encryption of y using cipher C, where
C \in {Caesar, Substitution, Morse, ...}. After decoding, the model is asked to execute the result. The process of encrypting y and having this timed-release mechanism is an analog of locking a message with a time-lock puzzle from our construction.

In total, we ran three types of experiments:
- Existing experiment in paper: Feed y to an input filter and if it passes, let an unaligned LLM answer the harmful prompt. Then feed the harmful answer to the output filter. We demonstrate that existing filters (Llama Guard 2 8B, ShieldGemma 2B) fail to detect that prompt y is dangerous, and that they are also unable to detect harmful outputs generated by unaligned LLMs (Llama 3.2 3B, Qwen 7B, Mistral 24B).
- Additional rebuttal experiment: Feed the timed-release attack z to an input filter (Llama Guard 2 8B, ShieldGemma 2B). Table A below shows that this timed-release mechanism achieves higher filter bypassing rates than directly feeding y to the input filters.
- Additional rebuttal experiment: Feed the timed-release attack z to a supposedly aligned LLM that handles alignment internally. We show that in some cases, Google Gemini 2.5 Flash refuses to answer on y but provides a harmful response for its encoding z, demonstrating that the timed-release attack can improve on existing jailbreaks.

The additional experiments b) and c) will be added to the updated version of the paper.

The experiments we run in b) and c) used a mechanism similar to that of a follow-up work [1] (the link leads to a paper that references our work in an anonymized way).
[1] designs an attack that successfully jailbreaks additional production models, e.g., Google Gemini (2.5 Flash/Pro), DeepSeek Chat (DeepThink), Grok (3), and Mistral Le Chat (Magistral). This attack is inspired, as explicitly said in [1], by our time-lock idea to hide harmful prompts y under some amount of computation. It shows that the alignment mechanisms embedded inside production models are not able to detect harmful commands hidden with a time-lock-like mechanism, but the models can recover the commands and produce harmful outputs. Importantly, the models given harmful prompts y in the clear (without the time-lock component) refuse to answer.


Table A:
# Input filter bypass rates by guard and encoding type

| Guard | No encryption | ASCII | Caesar | Morse | Substitution | Unicode |
|-------|---------------|-------|--------|-------|--------------|---------|
| Llama Guard 2 8B | 34.37% | 63.16% | 87.00% | 79.74% | 79.84% | 78.40% |
| ShieldGemma 2B | 79.49% | 78.40% | 91.89% | 87.15% | 91.98% | 87.97% |

[1] https://github.com/iclr18523/ControlledRelease

---

### Author Response · Authors · 2025-11-27
**Summary of revision**

We thank the reviewers again for their careful reading and helpful comments. We have substantially revised the paper to address the concerns raised. In the following we summarize how we changed the paper based on their general feedback:

**Structure of the paper:**
- Title: We shortened the title to “The Computational Cost of Filtering for AI Alignment"
- Structure: We restructured the paper with improved numbering to shorten the introduction and improve flow, and moved the related literature part to the main body.
- Figure: We removed the general figure in the introduction and instead added one in the technical overview of our construction to illustrate the time-lock puzzle.
- Philosophical perspective: We moved the philosophical perspective to the appendix.

**Explanations/Clarifications:**
- We made our contributions more explicit in the introduction, revised the section on input filtering (section 2, previously section 1.1) to be less dense, and made the overview of the construction (technical overview, section 7.1) more explicit with additional explanations.
- We also added clarifications to why it is realistic to assume that filters are more efficient than LLMs (see discussion, section 2).
- We added a new paragraph to our related literature section that directly compares the papers mentioned during the rebuttal to our paper.

**Experiments:**
- We added the new experiments (see rebuttal comments) to the experiments section (section 4), which support the practicability of our theoretical modelling and are more aligned with our theoretical construction as argued in the previous general comment. A detailed explanation of the experiments is added to the Appendix.

**Theorem 2:**
- We revised Theorem 2 (and Theorem 10 in the Appendix) to make it explicit on what prompts the LLM runs. We also made the theorem slightly more general by stating that it holds for every prompt to M.

We believe these revisions substantially address the reviewers' concerns while maintaining the paper's theoretical rigor and practical relevance. We address reviewer-specific concerns in our individual responses below.

---

### Author Response · Authors · 2025-12-02
**Rebuttal summary for AC**

Thank you for taking over the AC role. Quick rebuttal summary: You will see from the extensive discussions that our thorough responses and substantial improvements address the concerns raised. While name leakage prevented us from hearing back from all reviewers, we note that Reviewer 938F raised the score from 2 to 6 after reviewing our revisions. We expect that the other reviewers would have similarly acknowledged our rebuttal with an increased score. Below, we summarize what reviewers considered as the strengths of our paper, their concerns and how we addressed them.

**Strengths**\
The reviewers unanimously recognized the paper's significant theoretical contributions to AI safety, particularly valuing the novel use of computational complexity theory and cryptographic hardness assumptions (Time-Lock Puzzles, One-Way Functions) to formalize fundamental limitations of filtering-based alignment methods. They emphasized that this approach elevates the discussion "from empirical observation to theoretical impossibility under standard assumptions" with core separation theorems that are "robust and compelling" under realistic constraints reflecting practical black-box LLM deployments. Reviewers appreciated the comprehensive scope extending beyond simple detection to mitigation filters and various cryptographic scenarios. Overall, reviewers emphasized the originality and significance of our work, noting it addresses a "fundamental problem in AI safety" through a highly novel lens.

**Concerns and how we addressed them**\
**Assumptions**. Reviewers raised concerns about the efficiency assumption for filters not reflecting real-world safety decisions, and reliance on unproven cryptographic hardness assumptions. We addressed these by adding clarifications in Section 2 explaining why the efficiency assumption is realistic—guard models are typically much smaller than deployed LLMs (e.g. compare Llama Guard 4 (12B) to larger models like GPT-5) due to latency and cost constraints. Regarding cryptographic assumptions, we clarified that relying on well-tested conjectures is standard practice in cryptography, with real-world systems like internet private communication depending on similar assumptions.

**Clarity**. Reviewers found: i) Theorem 2 too informal, ii) the construction in the technical overview too abstract, and iii) Section 1.1 too dense. We revised Theorem 2 (and Theorem 10) to explicitly specify on what prompts the LLM runs and generalized it to hold for every prompt to M. We made our contributions more explicit in the introduction, rewrote the input filtering section (Section 2, before 1.1) to be more accessible, and enhanced the technical overview (Section 7.1) with additional explanations and a concrete figure illustrating the time-lock puzzle construction.

**Experiments**. Reviewers noted the lack of experimental support for our theoretical model and questioned practical relevance. In additional experiments we demonstrate that a mechanism inspired by our theory achieves significantly higher filter bypass rates (up to 91%) compared to direct jailbreak attempts, and successfully jailbreaks Gemini 2.5 Flash that refuses original jailbreak prompts. These results are included in Section 4 with details in the Appendix. Additionally, a follow-up work by independent researchers designed an attack that is inspired, as they explicitly say, by our time-lock idea to hide harmful prompts under some amount of computation. They successfully attack multiple systems (Gemini, DeepSeek, Grok, Mistral), providing strong empirical confirmation of our theoretical predictions.

**Structure**. We addressed concerns about structure by: shortening the title, restructuring with improved numbering to condense the introduction and improve flow, moving related work to the main body, rewriting the technical overview and replacing the abstract Figure 1 with a technical illustration of the time-lock puzzle in the construction overview. We also moved the philosophical discussion to the appendix to maintain focus on technical contributions.

**Scope**. Regarding the focus on external filters rather than embeddings, we discuss that extending our results to such filters is plausible. Our experiments on production models (which implicitly use intermediate embeddings for alignment) provide evidence that such extensions may be feasible, though formally proving this is outside the current scope.

**Related work**. We added a paragraph to the related literature section comparing our work to papers mentioned during rebuttal, clarifying how our provable, non-collusive approach based on time-lock puzzles differs from prior work on mosaic prompts (which require shared secrets) and cipher-based empirical jailbreaks (which lack theoretical guarantees or assumes joint secret between LLM and the attacker).

We view these revisions to substantially address the reviewers' concerns while maintaining theoretical rigor and demonstrating practical relevance.

---

### Meta-Review · Area_Chair_43yR · 2026-01-07

**Summary:**

This paper studies the mitigation of the potential misuse of LLMs in generating harmful content, with a focus on filters to prevent the generation of unsafe information. The paper discusses two challenges for filter-based defense: (1) there exist LLMs for which there are no efficient input prompt filters, and (2) identifies a setting in which output filtering is computationally intractable. Overall, the paper concludes that safety cannot be achieved by designing filters external to the LLM internals.

**Reviewer Concerns:**

Overall, the reviewers are positive (mostly with very low confidence) for the theoretical insights on filter-based defenses. The writing can be polished to make the paper more readable to the ICLR audience, and the relevance to the practice of the theory can be enhanced with additional experiments, e.g., experiments done during the rebuttal.

**Reviewer Scores:**

The reviewer 938F may increase their score to 6, and the other three reviewers may keep their scores.

---

### Decision · Program_Chairs · 2026-01-26

Accept (Poster)